# Integrated Single-Trait and Multi-Trait GWASs Reveal the Genetic Architecture of Internal Organ Weight in Pigs

**DOI:** 10.3390/ani13050808

**Published:** 2023-02-23

**Authors:** Xuehua Li, Jie Wu, Zhanwei Zhuang, Yong Ye, Shenping Zhou, Yibin Qiu, Donglin Ruan, Shiyuan Wang, Jie Yang, Zhenfang Wu, Gengyuan Cai, Enqin Zheng

**Affiliations:** 1College of Animal Science and National Engineering Research Center for Breeding Swine Industry, South China Agricultural University, Guangzhou 510642, China; 2Guangdong Provincial Key Laboratory of Agro-Animal Genomics and Molecular Breeding, South China Agricultural University, Guangzhou 510642, China; 3Yunfu Subcenter of Guangdong Laboratory for Lingnan Modern Agriculture, Yunfu 527400, China

**Keywords:** internal organ weight, GWAS, DLY pigs, genetic architecture

## Abstract

**Simple Summary:**

In this study, bioinformatics approaches were used to better understand the genetic architecture of internal organ weights in three-way crossbred commercial pigs and to map genetic markers and genes. For this purpose, we used single-trait and multi-trait genome-wide association studies (GWASs) followed by a haplotype block analysis. We explored the key genetic markers and genes from the internal organ weight genome-wide association study results of three-way crossbred commercial pigs. In this manner, five genes, *TPK1*, *POU6F2*, *PBX3*, *UNC5C*, and *BMPR1B*, were defined as central in affecting internal organ weight in pigs. Moreover, *APK1*, *ANO6*, and *UNC5C* were identified to be pleiotropic in multi-trait GWASs. These results can be applied to various types of genomic studies of pigs.

**Abstract:**

Internal organ weight is an essential indicator of growth status as it reflects the level of growth and development in pigs. However, the associated genetic architecture has not been well explored because phenotypes are difficult to obtain. Herein, we performed single-trait and multi-trait genome-wide association studies (GWASs) to map the genetic markers and genes associated with six internal organ weight traits (including heart weight, liver weight, spleen weight, lung weight, kidney weight, and stomach weight) in 1518 three-way crossbred commercial pigs. In summation, single-trait GWASs identified a total of 24 significant single- nucleotide polymorphisms (SNPs) and 5 promising candidate genes, namely, *TPK1*, *POU6F2*, *PBX3*, *UNC5C*, and *BMPR1B,* as being associated with the six internal organ weight traits analyzed. Multi-trait GWAS identified four SNPs with polymorphisms localized on the *APK1*, *ANO6*, and *UNC5C* genes and improved the statistical efficacy of single-trait GWASs. Furthermore, our study was the first to use GWASs to identify SNPs associated with stomach weight in pigs. In conclusion, our exploration of the genetic architecture of internal organ weights helps us better understand growth traits, and the key SNPs identified could play a potential role in animal breeding programs.

## 1. Introduction

Body weight, which can reflect growth performance and thus affect economic efficiency, has attracted a lot of attention in animal breeding programs. The body weight of cattle is the sum of various elements, including fat weight, internal organ weight, muscle weight, and bone weight, among others. Of these components, internal organ weight constitutes 14% of the total body weight of cattle [1]. The weight and size of an organ are salient features that serve as dependable predictors of its developmental progression, wherein an augmented organ mass typically alludes to a heightened degree of maturation. Accelerated organ development leads to a smoother coordination of internal organs during vital biological processes such as oxygen transport, blood circulation, lipid metabolism, and digestion. This refinement of these processes can positively impact growth and economic traits. Previous studies have shown that the internal organ weights of crossbred steer calves are strongly correlated with carcass growth rate [2]. Moreover, in humans, internal organ weights have been shown to be positively correlated with body weight and height in normal Zambian adults [3]. Thus, comprehending the genetic architecture of heart weight (Heart WT), liver weight (Liver WT), spleen weight (Spleen WT), lung weight (Lung WT), kidney weight (Kidney WT), and stomach weight (Stomach WT) will propel genetic progress and facilitate the successful implementation of breeding programs.

Genome-wide association studies (GWASs) are widely used to identify quantitative trait loci (QTL) and candidate genes associated with complex traits in animals and plants. To date, the number of QTL associated with Heart WT, Liver WT, Spleen WT, Lung WT, and Kidney WT are 29, 31, 19, 5 and 8, respectively, and no QTL have been reported to be associated with Stomach WT in the pig QTL database [4] (accessed on 15 November 2022). Previous studies reported 39 QTL to be associated with internal organ weight in four local pig populations and one commercial population [5]. For instance, Zhang et al. [6] showed that a 2 cM QTL on Sus scrofa chromosome 2 (SSC2) was significantly associated with Heart WT, and three QTL were associated with Liver WT, Lung WT, and Spleen WT. Although several studies have identified QTL to be associated with internal organ weight [7,8], the process of genetic improvement remains slow.

The difficulty (and high cost) of obtaining phenotypes for internal organ weight studies has led to fewer studies on its genetic architecture. Moreover, previous studies conducted single-trait GWASs for internal organ weight to map the genetic markers and genes; however, internal organ development is mutually coordinated by each different organ, and the single-nucleotide polymorphisms (SNPs) in the genome may act on multiple organs at the same time. Therefore, it is difficult to identify SNPs and candidate genes that affect multiple internal organs simultaneously using single-trait GWASs. Therefore, herein, we performed multi-trait GWASs to identify polymorphic SNPs and improve statistical efficiency, which mainly depends on the genetic correlation between traits [9,10]. In this manner, it was observed that the statistical efficiency was improved in the case of low trait correlations [11,12].

Previous studies demonstrated the superiority of conducting multi-trait GWASs in terms of uncovering the genetic architecture of complex traits in animals. For instance, Zhou et al. [13] performed multi-trait GWASs to identify 21 pleiotropic SNPs that were not detected via single-trait GWASs in three body size traits. In Simmental beef cattle, An et al. [14] detected 29 pleiotropic SNPs that were functional in all three growth periods using multi-trait GWASs. To date, there are no studies that use multi-trait GWASs to analyze the genetic architecture of visceral weight. Herein, we performed multi-trait GWASs to compensate for the deficiencies associated with single-trait GWASs and to provide new insights into the genetic mechanisms of multi-organ co-development.

The aim of this study was to map the genetic markers and candidate genes associated with internal organ weight in pigs. To this end, we conducted single-trait and multi-trait GWASs for six internal organ weight traits in 1518 crossbred commercial Duroc × (Landrace × Yorkshire) DLY pigs. The results from the current study advanced our understanding of the genetic basis for internal organ weight and further revealed the complexity of the genetic architecture of internal organ weight in pigs. Integrating SNP results from GWASs as a source of prior biological information in the improvement program enhances the selection process by assigning higher weight to key SNPs that are critical for improving internal organ weight traits.

## 2. Materials and Methods

### 2.1. Ethical Statement

All animals used in this study were treated in accordance with the guidelines for the use of laboratory animals of the Ministry of Agriculture of China and with the approval of South China Agricultural University (Guangzhou, China), No. 2018F089.

### 2.2. Animal Samples and Phenotype Collection

Experimental animals were selected from a DLY three-way crossbred commercial line with no overlapping blood relations, through random selection based on genealogy, in which 89 Duroc boars were mated with 397 Landrace × Yorkshire sows to produce a large number of offspring. All pigs were raised in four farms of the Guangdong Wens Food Group Co., Ltd. (Guangzhou, China). In brief, a total of 1518 individuals (757 boars and 764 sows) were reared with free access to water and feed and were fattened to 115 kg. They were euthanized in 13 batches with a 24 h interval between each batch and had an average slaughter age of about 7 months. After the pigs were euthanized, their phenotypes were recorded, and their internal organs were excised, emptied, flushed, blotted dry, and weighed immediately using an electronic scale with a range of 0.0 kg to 300 kg and accuracy of ±100 g. The scale was calibrated using the linear calibration method with 20% MAX or 60% MAX weight. The organ distribution is shown in Figure 1. R 4.2.1 software was used to test the normal distribution of the descriptive statistics of the internal organ traits.

### 2.3. Genotyping and Quality Control

Ear samples were collected from all 1518 individuals, and genomic DNA was extracted from the ear tissue of each pig using a standard phenol–chloroform method and subsequently diluted to 50 ng/μL for the genotyping procedure, controlling the quality OD260/280 between 1.8 and 2.0. The 1518 DLY pigs were genotyped using the GeneSeek Porcine 50K SNP BeadChip (Neogen, Lincoln, NE, USA), which contained 50,703 SNPs. After genotyping, to ensure the accuracy and validity of the GWAS results, we performed a quality control (QC) procedure using the PLINK v1.07 software [15] with the following parameters: individual call rate > 95%; SNP call rate > 99%; minor allele frequency > 1%; and *p* > 10^−6^ for the Hardy–Weinberg equilibrium test. Moreover, SNPs in sex chromosomes and unmapped regions were excluded. After QC, a final set of 31,941 eligible SNPs remained for subsequent single-trait and multi-trait GWASs.

### 2.4. Population Structure and Linkage Disequilibrium (LD) Estimation

PCA was conducted using the GCTA software [16] to assess the population structure, and PLINK v1.07 was used to calculate the LD decay distance, which was evaluated as the squared correlation of alleles (*r*^2^) with a window size of 1000.

### 2.5. Single-Trait and Multi-Trait Genome-Wide Association Studies

The GEMMA software [17] was used to implement the linear mixed model (LMM) for the single-trait GWAS of each internal organ weight trait, including heart weight, liver weight, spleen weight, lung weight, kidney weight, and stomach weight. GEMMA calculated the genomic relatedness matrix (GRM) between individuals to account for the population structure. The mixed linear model was as follows:y=Wα+Xβ+u+ε
where *y* is a vector of phenotypic values for each internal organ weight; *W* is the correlation matrix of covariates (fixed effects), including the top five eigenvectors of PCA, farm, sex, and slaughter lot; *α* is a vector of corresponding coefficients including the intercept; *X* is the genotypic vector of the SNP markers; *β* denotes the effect size of the SNP markers; *u* is a random effects vector, *u~MVNn (0, λτ^−1^K)*; *ε* is the residual vector, *ε~MVNn (0, ^τ−1^In)*; *λ* is the ratio of the specified variance components; *τ^−1^* is the variance of the residuals; *K* denotes the kinship matrix; *I* is the unit matrix; *n* is the number of individuals in the DLY population; *MVNn* denotes the multi-dimensional normal distribution.

Moreover, the GEMMA software [17] was used to implement the multivariate linear mixed models (mvLMMs) [18] for multi-trait GWASs to assess pleiotropic SNPs. The mvLMMs and LMMs were both implemented as described in previous studies [19]. In the current study, the LMMs and mvLMMs in the single-trait GWAS and the multi-trait GWAS utilized the same covariates. The multivariate linear mixed models were as follows:$$Y=WA+x\beta^{T}+U+E;\;G\;∼MN_{n\times d}\left(0,\;K,V_{g}\right),\;E\;∼MN_{n\times d}\left(0,I_{n\times n},V_{e}\right)$$
where Y is a matrix of six internal organs for 1518 individuals; W is a covariable matrix (fixed effects); A is a matrix of the corresponding coefficients; x is a vector that marks the genotypes; β is a vector of marker effect sizes for six internal organs’ weights. *U* denotes the random effects; E is a matrix of errors; K denotes the kinship matrix; $V_{g}$ denotes symmetric matrix of genetic variance component; *I* is an identity matrix; $V_{e}\;$denotes a symmetric matrix of the environmental variance component; $MN_{n\times d}\left(0,V_{1},V_{2}\right)$ denotes the n×d matrix normal distribution with mean 0; $V_{1}\;$denotes row covariance matrix; $V_{2}$ denotes column covariance matrix.

Furthermore, the Bonferroni correction can lead to an overcorrection and can be too conservative, this can result in a limited number of labeled association *p*-values that meet the standard across the genome. This can lead to a high false-negative rate. To address this issue, the false-discovery rate (FDR) was employed as a correction to the threshold [20]. Thus, the threshold *p*-value was calculated as $P=FDR*\frac{N}{M};$ the FDR was set to 0.01, *N* is the number of SNPs with *p*-value less than 0.01, and *M* refers to the total number of SNPs after quality control. Moreover, quantile–quantile (Q–Q) plots were constructed for the six internal organ weight traits to further assess the population structure.

In addition, the PLINK v1.07 and Haploview v4.2 software [21] were implemented to perform the haplotype block analysis in chromosomal regions with multiple significant SNPs. The default parameters of Haploview 4.2 [22] (MAF > 0.05, Mendelian error < 2, and *p*-value < 10^−3^ for the HWE test) were used to define the linkage disequilibrium (LD) blocks of SNPs.

### 2.6. Estimation of Heritability and Phenotypic Variation

In the present study, the restricted maximum likelihood (REML) method was used to assess the SNP-based heritability of each internal organ weight trait, and the percentage of phenotypic variation that could be explained by significant SNPs was calculated using GCTA software. SNP-based heritability and the percentage of phenotypic variation explained by significant SNPs were calculated as follows [23]:$$y=X\beta +g+\varepsilon \;with\;var\left(y\right)=A_{g}\sigma_{g}^{2}+I\sigma_{\varepsilon }^{2}$$
where *y* is the phenotypic value of each internal organ weight trait; *β* is the vector of fixed effects, including the top five eigenvectors of PCA, farm, sex, and slaughter lot; *X* is an association matrix; *g* is the vector of total genetic effect of all the qualified SNPs for the 1518 DLY pigs; $A_{g}$ is the genomic association matrix between different individuals; $\sigma_{g}^{2}$ is the additive genetic variance captured by either the genome-wide SNPs or the selected SNPs; $\sigma_{\varepsilon }^{2}$ refers to residual variance.

### 2.7. Candidate Gene Search and Function Analysis

Our previous studies on this population showed that the average *r*^2^ of 0.2 is about 200 kb apart [24]; the range for searching for the functional gene closest to the position of the significant SNP is determined based on the LD decay distance (*r*^2^ = 0.2) of the populations [25]. We used the “biomaRt” package [26] in R, based on the Sus scrofa 11.1 genome version database (http://ensemble.org/Sus_scrofa/Info/Index, accessed 20 September 2022). Genes nearest the significant SNPs are list in Tables. We conducted a search of both PubMed and the relevant literature to examine the correlation between the nearest peak SNPs of all the candidate genes and the internal organ weight traits being analyzed.

## 3. Results and Discussion

### 3.1. Phenotype Statistics and Heritability Estimation

The descriptive phenotypic statistics and estimated heritabilities (*h*^2^) for analysis of the internal organ weights are listed in Table 1. The weight of internal organs is a crucial indicator of internal organ development and has a significant impact on organ function. In the current study, the average Heart WT, Liver WT, Spleen WT, Lung WT, Kidney WT, and Stomach WT in DLY pigs were 455.57 g, 1763.61 g, 212.54 g, 1020.54 g, 0.41 kg, and 727.68 g, respectively. The estimated heritabilities of Heart WT and Lung WT were the lowest at 0.21 ± 0.04 and 0.28 ± 0.04, respectively, and all other organ weights had had moderate to high estimated heritabilities, ranging from 0.36 ± 0.04 to 0.49 ± 0.04. Similar to the results of a previous study, the estimated heritabilities of Heart WT, Liver WT, Spleen WT, and Kidney WT were between 0.35 and 0.54, which were moderate to high estimations [5], indicating that the estimated heritabilities of the weight of an internal organ is generally high in pigs and there is considerable room for improving the genetic contribution through breeding. Furthermore, the coefficients of variation were the lowest for Lung WT and all other traits were relatively high, indicating individual heterogeneity, low trait selection intensity, and high breeding potential.

Moreover, the genetic and phenotypic correlation coefficients among Heart WT, Liver WT, Spleen WT, Lung WT, Kidney WT, and Stomach WT are listed in Table 2. The results revealed moderate to low genetic correlations among the six internal organ weight traits. Heart WT had moderate genetic correlations with Liver WT, Lung WT, and Kidney WT, suggesting that these traits could be improved together in pig breeding programs. On the other hand, Stomach WT showed close to 0 genetic correlations with most of the other traits, indicating Stomach WT traits are less influenced by other traits when they are inherited. Therefore, reasonable breeding strategies need to be designed to improve internal organ weight traits. The phenotypic correlation results showed that the correlation coefficients between the phenotypes were at moderate to high levels, excluding the low phenotypic correlation coefficients between Lung WT and Liver WT, and Spleen WT and Kidney WT, especially the phenotypic correlation coefficients of Liver WT and Kidney WT were as high as 0.62. When selecting for a certain phenotype in pig breeding, it is advantageous to also consider other related traits.

### 3.2. Population Structure and LD decay

Population stratification is known to lead to false-positive results in GWASs. To detect potential population stratification, we performed PCA and added the first five principal components to the covariates of the GWAS model to correct for the population structure. Moreover, our previous study showed that the LD decay coefficient of the analyzed DLY pig population with *r*^2^ decayed to 0.2 at a physical distance of 200 kb [24], indicating that the DLY population is diverse with a weak linkage between loci, which facilitates the detection of key SNPs for internal organ weight traits. In addition, Q–Q plots were generated for Heart WT, Liver WT, Spleen WT, Lung WT, Kidney WT, and Stomach WT to further assess population stratification (together with the Manhattan plots: Figure 2). The expansion coefficients (lambda) of the Q–Q plots for all six internal organ weight traits were close to 1, and no overall systematic bias was observed, signifying a negligible effect of the DLY pig group structure on GWASs.

### 3.3. Single-Trait GWASs

Single-trait GWASs were performed for the weight of the heart, liver, spleen, lung, kidney, and stomach. The results showed that 6, 4, 3, 4, 3, and 4 SNPs were significantly associated with the weight of each organ, respectively. The results of these single-trait GWASs are presented in Figure 2 and Table 3. Notably, it is the first time that significant SNPs associated with Stomach WT have been identified in pigs. Furthermore, on the basis of the LD decay map, a region of 200 kb before and after the key SNPs was defined as a region to screen for candidate genes [24]. For heart weight, six significant SNPs were identified, located on SSC5, 6, 7, 12, and 14. These six SNPs surpassed the significance threshold of 1.01 × 10^−4^. Figure 2B shows an expansion coefficient lambda (λ) of 1.006. Details of the significant SNPs are listed in Table 3. The most significant SNP, WU_10.2_12_6703865 on SSC12, explains 1.70% of the phenotypic variation and is about 44 kb downstream of the *CD300LB* gene. The *CD300LB* gene is a triggering receptor expressed on bone marrow cells that regulates the cytosolic process of bone marrow cells [27], and the *CD300LB* protein stimulated by T cells regulates *DNMT3A* mutation and alters immune cells in heart failure [28].

For liver weight, four significant SNPs were detected on SSC4, SSC9, and SSC10 with a λ of 0.999 (Figure 2C,D and Table 3). These four SNPs surpassed the significance threshold of 1.04 × 10^−4^. The top SNP, H3GA0028070, accounted for 2.10% of the phenotypic variance and is located within the *TPK1* gene. A significant SNP, named ASGA0044340, 12 kb upstream of H3GA0028070, also located on *TPK1*, explained 0.82% of the phenotypic variation. According to reports, *TPK1* is a cofactor of certain enzymes associated with the glycolysis and energy production pathways. It is involved in the metabolism of water-soluble vitamins and cofactors and the thiamine metabolic pathway, and mutations in *TPK1* can cause thiamine metabolic dysfunction syndrome [29]. In addition, knockdown of this gene can lead to glycogen storage dysfunction [30]. However, no studies have shown *TPK1* to be directly associated with liver development and weight in pigs.

The GWAS results of Spleen WT identified three significant SNPs, located on SSC3, SSC9, and SSC18, with a λ of 0.972 (Figure 2E,F and Table 3). All three SNPs surpassed the threshold of significance (*p* < 1.20 × 10^−4^). The top SNP, ALGA0098928, explained 2.22% of the phenotypic variation and is located within *POU6F2*. *POU6F2* is a suppressor associated with nephroblastoma (WT) that regulates cell proliferation and specific differentiation [31]. According to the RT-qPCR results, the expression of *POU6F2* is associated with renal morphogenesis [32], suggesting that *POU6F2* may be closely associated with spleen weight traits.

For lung weight, four significant SNPs were detected on SSC1, SSC7, and SSC11 with a λ of 1.008 (Figure 2G,H and Table 3). These four SNPs surpassed the significance threshold of 1.04 × 10^−4^. An SNP named ALGA0110225 explained 1.81% of the phenotypic variance and is located 97 kb downstream of *PBX3*, indicating that *PBX3* and ALGA0110225 may both play a role in Lung WT. A literature review revealed that *PBX3* is directly regulated by targeting *NBPF10*, *miR-144*, and *miR-224*, which are directly associated with lung cancer cell proliferation [33]. In addition, overexpression of *PBX3* promotes the proliferation of A549 cells (lung cancer histiocytes) [34]. Therefore, we believe *PBX3* to be a promising candidate gene for influencing Lung WT, and the regulatory mechanism needs further investigation.

We performed GWASs with the Kidney WT trait in DLY pigs and detected three SNPs that were above the significance threshold (*p* < 1.03 × 10^−4^) (Figure 2I and Table 3). Figure 2J shows that the lambda is 0.987. WU_10.2_15_153747936 on SSC15 explains 1.97% of the phenotypic variation and is located 341 kb downstream of the *HDAC4* gene. Because the LD decay distance is 200 kb, it follows that *HDAC4* may not have a significant effect on Kidney WT traits.

QTL and significant SNPs have not been previously reported in relation to Stomach WT. Thus, this study is the first GWAS on pig Stomach WT. Herein, four significant SNPs were identified for the DLY pigs that were above the significance threshold of *p* < 1.17 × 10^−4^ (Figure 2K and Table 3). Of the four significant SNPs, three SNPs were simultaneously located on SSC8 and both MARC0052872 and ALGA0106192 were located within the *UNC5C* gene. Furthermore, the distance between MARC0052872 and ALGA0106192 was only 21 kb, explaining 2.53% and 2.62% of the phenotypic variation, respectively. ASGA0101191 was 100 kb away from the aforementioned SNPs with a phenotypic variation value of 2.14% and was located within *BMPR1B*. A literature review revealed that *UNC5C* plays a dominant role in netrin-1/*UNC5C*-mediated axonal rejection [35] and that its promoter region sequence binds to *p53* and acts as a target of *p53* to regulate apoptosis [36]. As regards the *BMPR1B* gene, it has been shown that the BMP family is expressed in the early organ and tissue formation during mouse embryonic development [37]. However, neither the *BMPR1B* nor the *UNC5C* gene is directly associated with internal organ weight traits.

The above GWAS results show that none of the SNPs associated with internal organ weight overlapped with those previously reported QTL documented in the pig QTL database [1]. This may have been due to the fact that most studies focused on the breeding of native Chinese pigs, and fewer studies were conducted on DLY three-way crossbred commercial populations with significant breed differences. Moreover, the significant SNPs did not overlap in the six traits, i.e., none of the SNPs were polymorphic, which may be related to the low genetic and phenotypic correlation between traits and the low density of genetic markers, which was further verified by multi-trait GWASs.

### 3.4. Haplotype Block Analysis

Figure 3 shows the LD pattern of significant SNPs associated with Stomach WT. In this study, multiple SNPs associated with Stomach WT were in close proximity to each other, with two significant SNPs on SSC8, which is located in a 21 kb region within the *UNC5C* and *BMPR1B* genes (the gene function is described above). The insufficient density of 50K microarray markers resulted in a low number of SNPs with linkage disequilibrium, which limited the resolution of the genetic architecture of key SNPs for the trait to some extent.

### 3.5. Multi-Trait GWASs

In order to improve the statistical effect, multi-trait GWASs were individually performed for each SNP by combining the joint analysis of six internal organ weight traits. This revealed the genetic factors with significant interactions among different traits in the same individual under the same environment. Manhattan plots of the multi-trait GWASs are shown in Figure 4.

The multi-trait GWASs combining six internal organ weight traits identified four significant SNPs with polymorphisms affecting the phenotypes, ALGA0032998, H3GA0028070, MARC0052872, and ALGA0106192 (Figure 5 and Table 4). SNP ALGA0032998 explained 1.36% of the phenotypic variation and is located within the *ANO6* gene. The overexpression of *CCR7* was observed to enhance the migration of BxPC-3 cells under the induction of the *ANO6* gene, which is a potential mediator of *ANO6* expression through the *ERK* signaling pathway. This promotion of migration was also seen in pancreatic ductal adenocarcinoma cells [38]. The single-trait GWAS described the effects of three SNPs (H3GA0028070, MARC0052872, and ALGA0106192) located on *TPK1* and *UNC5C* genes on the weight of the liver and stomach. These SNPs were not only significant in the single-trait GWAS but were also found to be simultaneously associated with the weight of all six internal organs, suggesting that these four SNPs have pleiotropic effects. Furthermore, no additional SNPs, independent of the single-trait GWAS results, were found. Similar results were previously reported by Guo et al. [39], in which no additional SNPs, independent of the single-trait GWAS results, were detected in the multi-trait GWASs for backfat thickness, carcass weight, and body weight in the DLY and Duroc populations. The reasons for this situation are manifold. For example, the complexity of the genetic architecture of the internal organ weight trait and the low marker density result in a low number of SNPs reaching significant levels. This renders LD detection insufficient and increases the difficulty of screening for co-dominant SNP or QTL regions. Thus, a larger sample population and a higher marker density are required to screen for loci associated with internal organ weight.

## 4. Conclusions

In this study, we conducted single-trait and multi-trait GWASs on the internal organ weights of 1518 DLY pigs. A total of 24 significant SNPs were detected in the single-trait GWAS results for six internal organ weight traits. The four significant pleiotropic SNPs identified via multi-trait GWASs were associated with six internal organ weight traits, confirming the results of the single-trait GWASs and improving our ability to reveal the genetic architecture of organ weight traits. *TPK1*, *POU6F2*, *PBX3*, *UNC5C*, and *BMPR1B* were highlighted as potential genes responsible for differences in Liver WT, Spleen WT, Lung WT, and Stomach WT among individuals according to their gene functions. In summary, the results of this study contribute to our understanding of the genetics of internal organ weight traits in DLY pigs by assigning higher weights to relevant SNPs and key genes in the genome.

## Figures and Tables

**Figure 1 animals-13-00808-f001:**
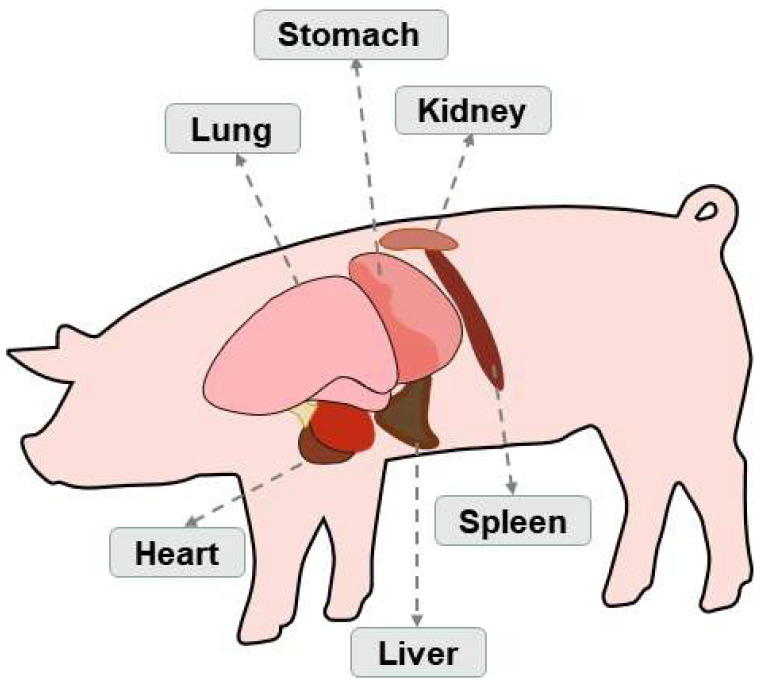
Distribution of heart, liver, spleen, lung, kidney, and stomach in the pig.

**Figure 2 animals-13-00808-f002:**
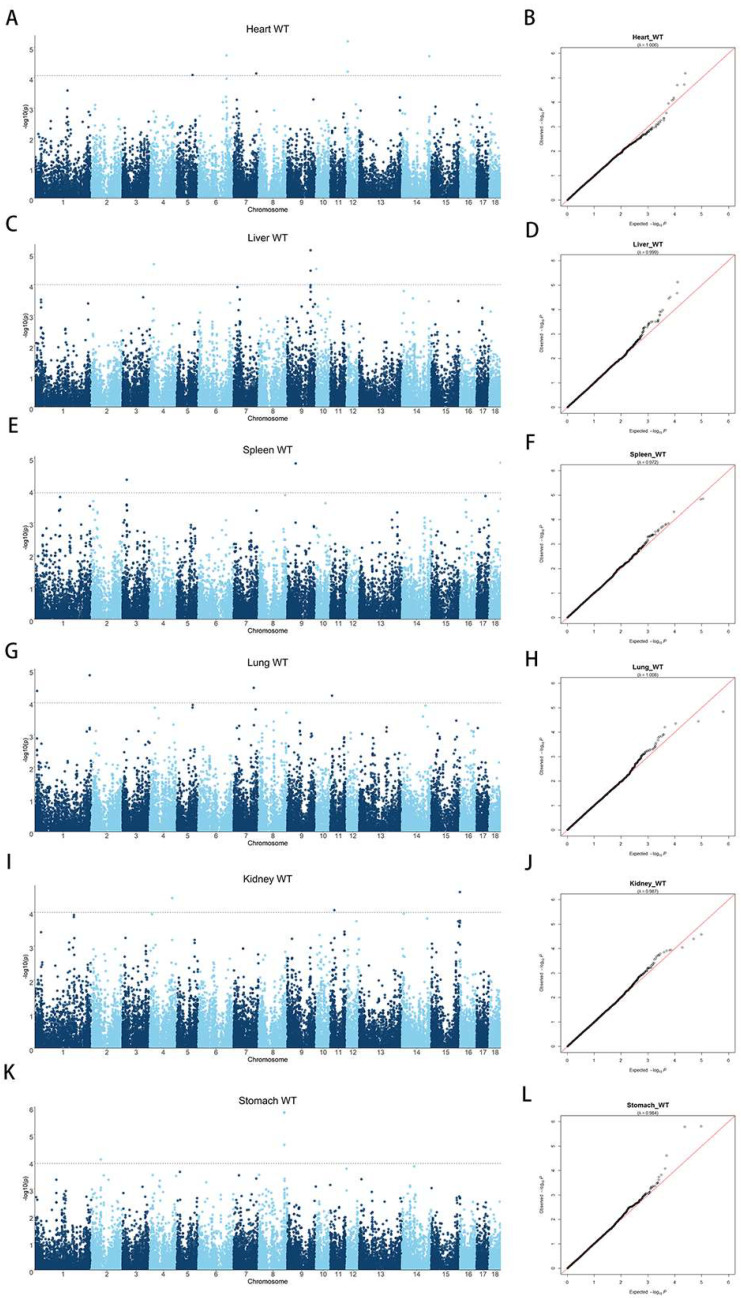
Manhattan and Q–Q plots of internal organ weight traits in the single-trait GWAS. (**A**) GWAS for Heart WT; (**B**) Q–Q plot for Heart WT; (**C**) GWAS for Liver WT; (**D**) Q–Q plot for Liver WT; (**E**) GWAS for Spleen WT; (**F**) Q–Q plot for Spleen WT; (**G**) GWAS for Lung WT; (**H**) Q–Q plot for Lung WT; (**I**) GWAS for Kidney WT; (**J**) Q–Q plot for Heart WT; (**K**) GWAS for Stomach WT; (**L**) Q–Q plot for Stomach WT. The *x*-axis represents the chromosome, and the *y*-axis represents the −log10 (*p*-value) value in the Manhattan plot of the GWAS. The Q–Q plot is plotted with the *x*-axis representing the actual measured value of −log10 (*p*-value) and the *y*-axis representing the observed value of −log10 (*p*-value) and labeled with the expansion factor lambda (λ).

**Figure 3 animals-13-00808-f003:**
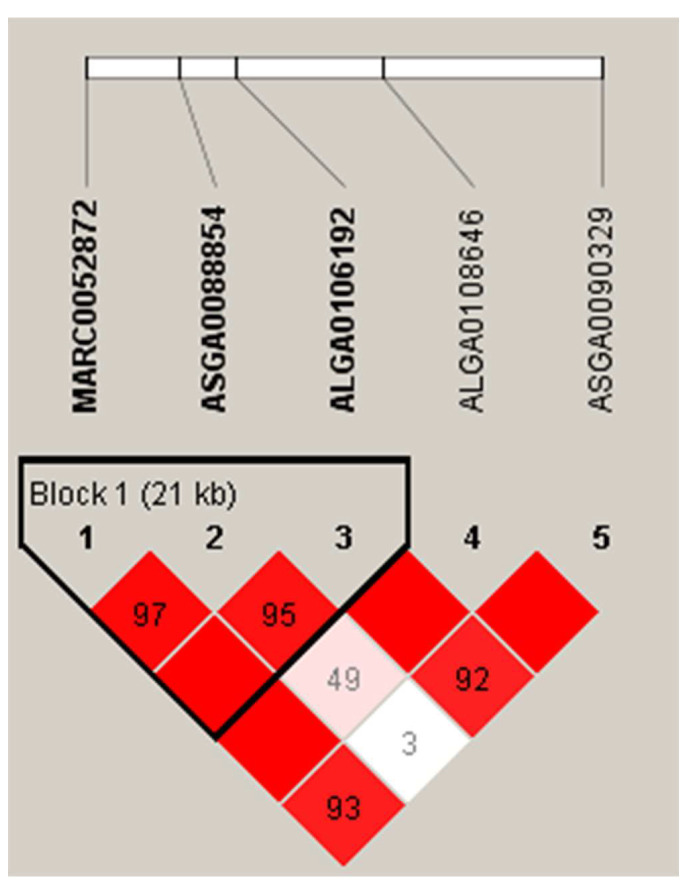
Linkage disequilibrium blocks of important SNPs on SSC8 in DLY pigs. The value in the box is the degree of linkage disequilibrium between SNPs (*r^2^*).

**Figure 4 animals-13-00808-f004:**
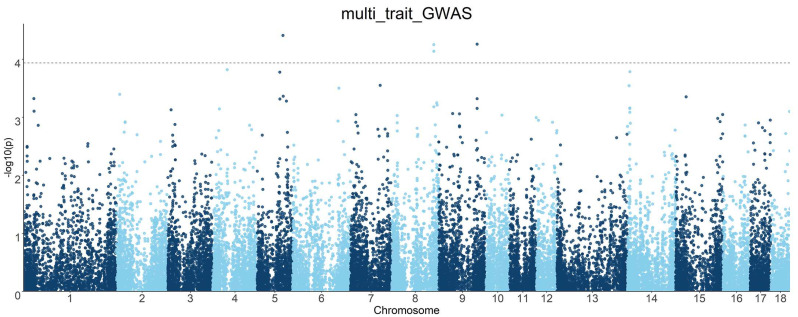
Manhattan plots of multi-trait GWAS results for six internal organ weight traits in the DLY population. The *x*-axis represents the chromosome, and the *y*-axis represents the −log10 (*p*-value) value in the Manhattan plot of the GWAS. The dashed line indicates the FDR-corrected threshold.

**Figure 5 animals-13-00808-f005:**
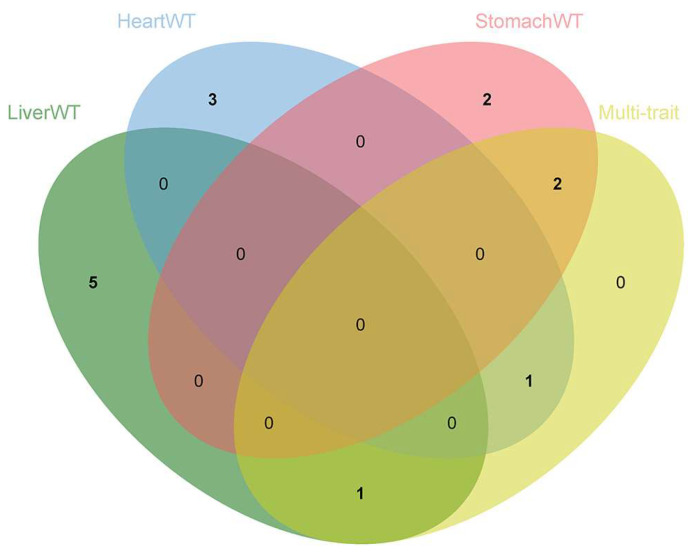
Venn diagram showing the distribution of SNPs in the single-trait GWASs and multi-trait GWASs, highlighting the role of pleiotropic SNPs in multiple traits.

**Table 1 animals-13-00808-t001:** Phenotypic statistics and heritability estimates for Heart WT, Liver WT, Spleen WT, Lung WT, Kidney WT, and Stomach WT.

Trait	N	Mean (±SD)	Min	Max	C.V.% ^a^	*h*^2^ (±SE)
Heart WT	1518	455.57 ± 78.06	217.1	868.8	17.13	0.21 ± 0.04
Liver WT	1518	1763.61 ± 271.83	993.9	2607.7	15.41	0.46 ± 0.04
Spleen WT	1517	212.54 ± 47.49	95.9	502.1	22.34	0.49 ± 0.04
Lung WT	1517	1020.54 ± 240.10	918.9	2090.4	2.18	0.28 ± 0.04
Kidney WT	1486	0.41 ± 0.08	0.17	0.84	19.51	0.36 ± 0.04
Stomach WT	1518	727.68 ± 129.97	495.7	1321.7	17.86	0.47 ± 0.04

^a^ Coefficient of variation (C.V.).

**Table 2 animals-13-00808-t002:** Phenotypic correlations (above the diagonal) and genetic correlations (below the diagonal) among organ weight traits within the DLY population.

	Heart WT	Liver WT	Spleen WT	Lung WT	Kidney WT	Stomach WT
Heart WT	1	0.36	0.33	0.41	0.37	0.49
Liver WT	0.31 ± 0.11	1	0.34	−0.02	0.62	0.38
Spleen WT	0.23 ± 0.11	0.11 ± 0.09	1	0.15	0.33	0.41
Lung WT	0.33 ± 0.13	0.17 ± 0.11	0.04 ± 0.11	1	0.05	0.27
Kidney WT	0.30 ± 012	0.33 ± 0.09	0.07 ± 0.10	0.29 ± 0.12	1	0.43
Stomach WT	0.02 ± 0.12	0.03 ± 0.09	0.26 ± 0.09	0.02 ± 0.11	0.03 ± 0.10	1

**Table 3 animals-13-00808-t003:** Significant SNPs and candidate genes for Heart WT, Liver WT, Spleen WT, Lung WT, Kidney WT, and Stomach WT in single-trait GWASs.

Trait	SSC	SNP	Position (bp)	MAF	*p*-Value	PEV (%) ^a^	Candidate Gene	Distance
Heart WT	6	WU_10.2_6_126961053	137,008,535	0.257	1.88 × 10^−5^	2.09%	*ST6GALNAC3*	Within
	14	6_43731895	132,228,312	0.407	1.99 × 10^−5^	1.71%	*HTRA1*	124,717
	12	WU_10.2_12_6703865	6,682,110	0.389	6.40 × 10^−5^	1.70%	*CD300LB*	44,143
	5	ALGA0032998	76,317,972	0.264	8.30 × 10^−5^	1.36%	*ANO6*	Within
	7	WU_10.2_7_116585612	110,088,932	0.242	7.42 × 10^−5^	1.10%	*KCNK10*	−29,135
	12	12_5381300	5,426,010	0.319	6.50 × 10^−5^	0.53%	*CDK3*	Within
Liver WT	4	WU_10.2_4_20570494	19,550,555	0.249	2.15 × 10^−5^	2.11%	*CCN3*	−35,758
	9	H3GA0028070	113,152,352	0.09	7.54 × 10^−5^	2.10%	*TPK1*	Within
	9	ASGA0044340	113,140,343	0.118	3.51 × 10^−5^	0.82%	*TPK1*	Within
	10	WU_10.2_10_3469625	1,753,591	0.476	3.08 × 10^−5^	0.43%	*RGS21*	Within
Spleen WT	18	ALGA0098928	54,993,603	0.328	1.39 × 10^−5^	2.22%	*POU6F2*	Within
	3	ALGA0105765	20,525,651	0.216	4.74 × 10^−5^	0.78%	*HS3ST4*	−19,582
	9	WU_10.2_9_45824613	40,989,995	0.399	1.47 × 10^−5^	0.58%	*TTC12*	Within
Lung WT	11	ALGA0060656	8,759,687	0.231	6.24 × 10^−5^	2.76%	*FRY*	Within
	1	ALGA0110225	266,708,292	0.278	1.46 × 10^−5^	1.81%	*PBX3*	97,646
	7	ASGA0035515	98,022,168	0.04	3.57 × 10^−5^	1.43%	*YLPM1*	Within
	1	MARC0089438	11,012,474	0.271	4.46 × 10^−5^	0.69%	*/*	/
Kidney WT	15	WU_10.2_15_153747936	138,955,316	0.369	2.61 × 10^−5^	1.97%	*/*	/
	4	WU_10.2_4_119054114	108,872,194	0.478	3.93 × 10^−5^	1.62%	*RAP1A*	122,469
	11	WU_10.2_11_20231427	19,917,312	0.221	8.86 × 10^−5^	0.88%	*SUCLA2*	Within
Stomach WT	8	ALGA0106192	124,443,641	0.064	1.54 × 10^−5^	2.62%	*UNC5C*	Within
	8	MARC0052872	124,421,940	0.063	1.61 × 10^−5^	2.53%	*UNC5C*	Within
	8	ASGA0101191	124,548,240	0.081	2.40 × 10^−5^	2.14%	*BMPR1B*	Within
	2	MARC0018316	46,014,239	0.46	8.35 × 10^−5^	0.74%	*ARNTL*	Within

^a ^The percentage of phenotypic variance explained by each SNP.

**Table 4 animals-13-00808-t004:** Significant SNPs and candidate genes for Heart WT, Liver WT, Spleen WT, Lung WT, Kidney WT, and Stomach WT in multi-trait GWASs.

SSC	SNP	Position (bp)	MAF	*p*-Value	PEV (%) ^a^	Candidate Gene	Distance
8	ALGA0106192	124,443,641	0.064	7.29 × 10^−5^	2.62%	*UNC5C*	Within
8	MARC0052872	124,421,940	0.063	5.61 × 10^−5^	2.53%	*UNC5C*	Within
9	H3GA0028070	113,152,352	0.088	5.51 × 10^−5^	2.10%	*TPK1*	Within
5	ALGA0032998	76,317,972	0.266	3.88 × 10^−5^	1.36%	*ANO6*	Within

**^a^** The percentage of phenotypic variance explained by each SNP.

## Data Availability

Data are contained within the article.

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
