# Peer review of "Integrated Single-Trait and Multi-Trait GWASs Reveal the Genetic Architecture of Internal Organ Weight in Pigs"

_animals, 2023, doi:10.3390/ani13050808_

Round 1

Reviewer 1 Report

Manuscript ID animals-2179096 entitled “Integrated Single-trait and Multi-trait GWASs Reveal the Genetic Architecture of Internal Organ Weight in Pigs”.  

 GENERAL COMMENTS: 

This work provides new information to the body of work. The results of this study are interesting and provide new insight to the scientific community. Please provide more details about how these traits could be added into selection improvement programs. Is the goal to increase/decrease the size of the organs? To speed up/slow down organ development? It is unclear how these traits could be added to a selection improvement program and how the contribute to economic traits. Overall, a significant review of grammar, sentence structure and proper English is recommended before publication.

ABSTRACT:  

L30: Change “In sum” to “In summation”

INTRODUCTION: 

L48: “Previous study has” should be “Previous studies have”

MATERIALS AND METHODS:  

Section 2.2: please add additional information regarding the number of males vs females in this study, the length of time between batches and the average age at slaughter. Please list the six internal organs that were weighed in the study, what equipment was used in weighing and how the equipment was calibrated. Additionally, please add if the pigs were randomly selected or if there were litter mates used. If litter mates were used, please describe details on number of littermates per litter.

RESULTS AND DISCUSSION:

L180: add respectively after the list of numbers

L181: add standard errors to the heritability values

L181 and 184: same comment as above

L217 – 220: Extreme run on sentence, please re-write. Add respectively after the list of numbers.

L253: change to “SNP name ALGA0110225 explained 1.81% of the phenotypic variance …” as it is currently written does not make sense.

L315-322: these sentences belong in the introduction not in the results/discussion

L324: what phenotype are you referring to here? It is not clear. Please also list the SNP that were significant in the multi trait GWAS in the text. A short discussion on those SNP/potential QTL related to the significant SNP would be beneficial in this section

TABLES AND FIGURES:  

Table 2: Please add standard error/deviation to the phenotypic correlations

Author Response

Response to Reviewer 1 Comments

Dear Editor and Reviewers,

On behalf of my co-authors, we thank you very much for giving us an opportunity to revise our manuscript. We appreciate editor and reviewers very much for their positive and constructive comments and suggestions on our manuscript entitled ”Integrated Single-trait and Multi-trait GWASs Reveal the Genetic Architecture of Internal Organ Weight in Pigs” (animals-2179096). Their suggestions have enabled us to improve our work. Appended to this letter is our point-by-point response to the comments raised by the reviewers.

We believe we have adequately addressed all comments, which we hope you will find satisfactory.

Best wishes,

Enqin Zheng

National Engineering Research Center for Breeding Swine Industry, South China Agricultural University

Guangzhou, Guangdong Province, 510642, P.R. China

Reviewer 1:

Here are the general comments from the reviewer: This work provides new information to the body of work. The results of this study are interesting and provide new insight to the scientific community. Please provide more details about how these traits could be added into selection improvement programs. Is the goal to increase/decrease the size of the organs? To speed up/slow down organ development? It is unclear how these traits could be added to a selection improvement program and how the contribute to economic traits. Overall, a significant review of grammar, sentence structure and proper English is recommended before publication.

Response: We thank the reviewer for the very meaningful comment. To be more clear and in accordance with the reviewer concerns, we have added a brief description as follows :

  • We add some sentences “Integrating SNP results from GWAS as a source of prior biological information in the improvement program enhances the selection process by assigning higher weight to key SNPs that are critical for improving internal organ weight traits.” to provide more details about how these traits could be added into selection improvement programs. Please see page 3 Lines 103-106.
  • I hold the view that there exists a correlation between organ weight and size, where heavier organs tend to be larger in size. These two factors serve as clear indicators of an organ's developmental stage. Our objective is to accelerate organ development, which would result in a better coordination of internal organs during vital biological processes such as oxygen transport, blood circulation, lipid metabolism, and digestion. This refinement of these processes can positively impact growth and economic traits. We have changed the sentence to “The weight and size of an organ are salient features that serve as dependable predictors of its developmental progression, wherein an augmented organ mass typically alludes to a heightened degree of maturation. Accelerated organ development leads to a smoother coordination of internal organs during vital biological processes like oxygen transport, blood circulation, lipid metabolism, and digestion. This refinement of these processes can positively impact growth and economic traits.” Please see page 2 Lines 46-52. We have changed sentences “Therefore, understanding the genetic architecture of heart weight (Heart WT), liver weight (Liver WT), spleen weight (Spleen WT), lung weight (Lung WT), kidney weight (Kidney WT), and stomach weight (Stomach WT) will advance the genetic progress and improve the value of economic traits which further contribute to the effective implementation of breeding programs.” to “Thus, comprehending the genetic architecture of heart weight (Heart WT), liver weight (Liver WT), spleen weight (Spleen WT), lung weight (Lung WT), kidney weight (Kidney WT), and stomach weight (Stomach WT) will propel genetic progress and facilitate the successful implementation of breeding programs.”in the revised manuscript. Please see page 2 Lines 57-61.
  • In the revised manuscript, we have completed the professional English editing by MDPI Editing Services.

L30: Change “In sum” to “In summation”

Response: The text has been modified according to the reviewer’s suggestion. Please see page 1 Line 29.

INTRODUCTION:

L48: “Previous study has” should be “Previous studies have”

Response: The text has been modified according to the reviewer’s suggestion. Please see page 1 Line 54.

MATERIALS AND METHODS: 

Section 2.2: please add additional information regarding the number of males vs females in this study, the length of time between batches and the average age at slaughter. Please list the six internal organs that were weighed in the study, what equipment was used in weighing and how the equipment was calibrated. Additionally, please add if the pigs were randomly selected or if there were litter mates used. If litter mates were used, please describe details on number of littermates per litter.

Response: We have revised the manuscript following the reviewer’s comments.. In the revised manuscript, we add the some information, i.e.,: " Experimental animals were selected from a DLY three-way crossbred commercial line with no overlapping blood relations, through random selection based on genealogy., in which 89 Duroc boars were mated with 397 Landrace × Yorkshire sows to produce a large number of offspring. All pigs were raised in four farms of Guangdong Wens Food Group Co., Ltd (Guangzhou, China). In brief, a total of 1518 individuals (757 boars and 764 sows) were reared with free access to water and feed and were fattened to 115 kg. They were euthanized in 13 batches with a 24-hour interval between each batch and had an average slaughter age of about 7 months. After the pigs were euthanized, their phenotypes were recorded and their internal organs were excised, emptied, flushed, blotted dry, and weighed immediately using an electronic scale with a range of 0.0 kg to 300 kg and accuracy of ± 100 g. The scale was calibrated using the linear calibration method with 20%MAX or 60%MAX weight. The organ distribution is shown in Figure 1. R software was used to test the normal distribution of the descriptive statistics of the internal organ traits.” Please see page 3 Lines 113-127.

RESULTS AND DISCUSSION:

L180: add respectively after the list of numbers

Response: Thanks for your reminding. We have added “respectively” after the list of numbers. Please see page 6 Line 233.

L181: add standard errors to the heritability values

Response: Thanks for your reminding. We have added standard errors to the heritability values. Please see page 6 Lines 234-236.

L181 and 184: same comment as above

Response: Thanks for your reminding. We have added standard errors to the heritability values. The heritability values in “Line 238” are cited from the reference literature, and the reference does not include the standard error. Please see page 6 Lines 234-236.

L217 – 220: Extreme run on sentence, please re-write. Add respectively after the list of numbers.

Response:

  • Thanks for your comment. This sentence we have rewritten to “Single-trait GWAS were performed for the weight of the heart, liver, spleen, lung, kidney, and stomach. The results showed that 6, 4, 3, 4, 3, and 4 SNPs were significantly associated with the weight of each organ, respectively. The results of this single-trait GWAS are presented in Figure 2 and Table 3” in the revised manuscript
  • We have added “respectively” after the list of numbers. Please see page 7 Lines 283-286.

L253: change to “SNP name ALGA0110225 explained 1.81% of the phenotypic variance …” as it is currently written does not make sense.

Response: Thanks for your helpful suggestion. We have changed the sentence to “SNP name ALGA0110225 explained 1.81% of the phenotypic variance and is located 97 kb downstream of PBX3, indicating PBX3 and AL-GA0110225 may both play a role in the Lung WT.” in the revised manuscript Please see page 8 Lines 324-325.

L315-322: these sentences belong in the introduction not in the results/discussion

Response: Thanks for your helpful suggestion. We excluded these sentences because they duplicated the information contained in section 1 " Previous studies have shown the superiority of conducting multi-trait GWAS in terms of uncovering the genetic architecture of complex traits in animals. For instance, Zhou et al. [13] performed multi-trait GWAS to identify 21 pleiotropic SNPs that was not detected by the single-trait GWAS in three body size traits. In Simmental beef cattle, An et al. [14] detected 29 pleiotropic SNPs that were functional in all three growth periods using a multi-trait GWAS.".Please see page 12 Line 391.

L324: what phenotype are you referring to here? It is not clear. Please also list the SNP that were significant in the multi trait GWAS in the text. A short discussion on those SNP/potential QTL related to the significant SNP would be beneficial in this section

Response:

  • We are sorry for the ambiguous described. We have changed the sentence “the results confirmed the significant SNPs identified in single-trait GWAS, located on ANO6, TPK1, and UNC5C genes.” to “The single-trait GWAS described the effects of three SNPs (H3GA0028070, MARC0052872, and ALGA0106192) located on TPK1 and UNC5C genes on the weight of the liver and stomach.” in the revised manuscript Please see page 12 Lines 406-408.
  • Thank you for your helpful suggestion. We added significant SNPs in the multi-trait GWAS in the revised manuscript. Please see page 12 Line 400.
  • We add some sentences “SNP name ALGA0032998 explained 1.36% of the phenotypic variation located within the ANO6 gene. The overexpression of CCR7 was observed to enhance the migration of BxPC-3 cells under the induction of the ANO6 gene, which is a potential mediator of ANO6 expression through the ERK signaling pathway. This promotion of migration was also seen in pancreatic ductal adenocarcinoma cells [38]. ” to discuss the significant SNPs in multi-trait and single-trait GWAS. Please see page 12 Lines 401-406.

TABLES AND FIGURES: 

Table 2: Please add standard error/deviation to the phenotypic correlations.

Response: We are very sorry for this suggestion. In this study, calculated the phenotypic correlation (rp) of the sample via Pearson's correlation coefficient using SPSS software, and the formula for calculating the Pearson's correlation coefficient is given by

where X and Y refer to the two traits; rp is Pearson's correlation coefficient between the two traits; n represents the number of samples; Xi and Yi are the phenotype values of X and Y traits for the i th individual, respectively;  and  refer to the average phenotype values of the X and Y traits phenotype across all samples, respectively.

The Pearson correlation coefficient is an indicator used to assess the degree of linear correlation between two variables, but it does not have the concept of standard deviation. So we are very sorry that we have no way to provide the standard error/deviation of the phenotypic correlation. (Table 2 Line 262)

Reviewer 2 Report

The authors presented single-trait and multi-trait GWAS analyses for six internal organ weight traits in crossbred commercial pigs. They highlight that this is the first GWAS study on stomach weight in pigs which makes the study original. The analyses seem appropriate and results and discussion section is well presented. The limitation of this study is the medium density SNP panel with reduction in the number of SNP after QC. I see the number of animals used in the GWAS as a limitation as well. The two facts do not invalidate the publication of the manuscript in a scientific journal.

Introduction

Line 46: “with internal organ weight accounting for 14%”. Specify the species.

Line 47: “operationally, jointly”. Correct text.

Line 58: Replace “To date, the identified QTLs associated” for “To date, the number of QTLs associated”.

Line 63: “SSC2”. Maybe replace for “on Sus scrofa chromosome 2 (SSC2)”

Line 63: exclude the parenthesis.

Paragraph 2: I would suggest breaking this paragraph into two or more. Start a new paragraph before “The difficulty”. Start a new paragraph before “Previous studies have shown”.

Comments on this section: some ideas are repetitive in the text. An edition to even reduce and refine the text would be valuable in this sense.

M&M

Line 103: Correct sentence. “Test for normal distribution…”

Comments on section 2.2: It is important to describe how these measurements were taken. The procedures.

Lines 114-118: replace “with standard:” for “with the following parameters:” Replace “filtered” for “excluded”. Exclude “still”. Replace “single-trait GWAS and multi-trait GWAS” for “single-trait and multi-trait GWAS”.

Lines 138-142: Provide more details on the multivariate linear models or specify the one that the authors chose to conduct the analysis.

Line 144: number of false negative results or false positive results? I believe false positive is more appropriate.

Line 152: “MAF values > 0.05”; “P-values <10³”. Check parameters because the authors were stricter in the genotyping quality control.

Line 160: “victor”. Typo.

Section 2.6: Describe how the SNP-based heritability was computed. Provide a reference for that as well.

Line 171: “in involved”. Text correction need.

Section 2.7: “nearest peak SNPs”. What was the window size established? Specify that in text. Usually we use biomaRt for that, but it seems the authors went to a manual search, right? I have found information in the Results section, but I believe this search and why the use of 200kb should be better explained in the M&M.

Results and Discussion

Lines 175-180. Text edition suggestion: The descriptive phenotypic statistics and estimated heritabilities (h²) for analyzed of the internal organ weights are listed in Table 1.

Second sentence needs correction. Please correct it. “For internal organ weights, which were important parameter responding organ function and the level of individual development.

Text edition suggestion: The average weight of heart WT, liver WT,spleen WT, lung WT, kidney WT and stomach WT in DLY pigs were 455.57g, 1763.61g, 212.54g, 1020.54g, 0.41kg, and 727.68g, respectively.

Exclude the next “for DLY pigs” in line 181.

Line 181: replace “had moderate” for “had moderate to high”.

Line 184: replace “were also moderate heritability” for “were moderate to high estimations”.

Line 195: result -> results

Line 195-200: “The results… improve the organ weight traits”. These is a long sentence. Try to break it down into two or more sentences.

Line 197: What does a “significant genetic independence” mean? The term significant is not appropriate here.

Line 196: “generally low”. Some moderate correlation coefficients were estimated. There is room for discussing the correlation results better. Value 0.62 for liver WT and kidney WT stood out.

Line 211: Please, show Q-Q plots in the manuscript or in the supplementary file. Now I see Fig 2 should be refereed in this section.

Line 227: and nearby -> and is located nearby. What is the distance between SNP 226 WU_10.2_12_6703865 and gene CD300LB in bp? Include this information in the manuscript.

Line 235: and within -> and is located within

Line 246: variation is -> variation and is

Line 254: that both vs may both play. Please, verify the text.

Line 282: reported by pigQTL database-> reported QTL documented in the pigQTL database. Include reference for pigQTL database.

Figure 2: replace “Manhattan and Q-Q plots of internal organ weight traits” for “Manhattan and Q-Q plots of internal organ weight traits in the single-trait GWAS”.

Table 3: Sort column SSC within trait to organize this table.  OR sort the table by the column “p-value”  or PEV from the highest to the lowest value. The same works for Table 4.

3.4. Multi-trait GWAS

Lines 312-315: Specify methodology for multi-trait GWAS in M&M and cite the reference.

Line 337: “internal organ-heavy traits”?

Line 316: “Populus trichocarpa” in italic.

Line 321: “All these results indicated that multi-trait GWAS supplemented the 321 results of single-trait GWAS and improved the statistical effect.” I believe this statement can be explained more because of the methodology implemented than the results found.

Conclusion

Replace “GWASs” with GWAS.

Author Response

Response to Reviewer 2 Comments

Dear Editor and Reviewers,

On behalf of my co-authors, we thank you very much for giving us an opportunity to revise our manuscript. We appreciate editor and reviewers very much for their positive and constructive comments and suggestions on our manuscript entitled ”Integrated Single-trait and Multi-trait GWASs Reveal the Genetic Architecture of Internal Organ Weight in Pigs” (animals-2179096). Their suggestions have enabled us to improve our work. Appended to this letter is our point-by-point response to the comments raised by the reviewers.

We believe we have adequately addressed all comments, which we hope you will find satisfactory.

Best wishes,

Enqin Zheng

National Engineering Research Center for Breeding Swine Industry, South China Agricultural University

Guangzhou, Guangdong Province, 510642, P.R. China

Reviewer 2:

The authors presented single-trait and multi-trait GWAS analyses for six internal organ weight traits in crossbred commercial pigs. They highlight that this is the first GWAS study on stomach weight in pigs which makes the study original. The analyses seem appropriate and results and discussion section is well presented. The limitation of this study is the medium density SNP panel with reduction in the number of SNP after QC. I see the number of animals used in the GWAS as a limitation as well. The two facts do not invalidate the publication of the manuscript in a scientific journal.

Response: We thank the reviewer for the enthusiasm expressed. We have revised the manuscript following the reviewers’ comments. Point-by-point responses to these comments are given below. Changes made to the text have also been marked using the "Track Changes" function.

Introduction

Line 46: “with internal organ weight accounting for 14%”. Specify the species.

Response: Thanks for your comment. We have added the species in the revision, “Body weight of cattle is the sum of various elements, including fat weight, internal or-gan weight, muscle weight, and bone weight, among others. Of these components, internal organ weight constitutes 14% of the total body weight of cattle.” Please see Page 2 Lines 43-46.

Line 47: “operationally, jointly”. Correct text.

Response: Corrected

Line 58: Replace “To date, the identified QTLs associated” for “To date, the number of QTLs associated”.

Response: The text has been modified according to the reviewer’s suggestion. Please see Page 2 Line 69.

Line 63: “SSC2”. Maybe replace for “on Sus scrofa chromosome 2 (SSC2)”

Response: The text has been modified according to the reviewer’s suggestion. Please see Page 2 Line 74.

Line 63: exclude the parenthesis.

Response: The text has been modified according to the reviewer’s suggestion. Please see Page 2 Line 74.

Paragraph 2: I would suggest breaking this paragraph into two or more. Start a new paragraph before “The difficulty”. Start a new paragraph before “Previous studies have shown”.

Response: Thank you for your helpful comments. We have broken paragraph 2 into three paragraphs in the revised manuscript. Please see Pages 2-3.

Comments on this section: some ideas are repetitive in the text. An edition to even reduce and refine the text would be valuable in this sense.

Response: Thanks for your helpful suggestion.

  • We have changed the sentence ”Internal organs coordinate with each other operationally, jointly regulating the growth and development of the whole organism, indirectly affecting production profits.” to “The weight and size of an organ are salient features that serve as dependable predic-tors of its developmental progression, wherein an augmented organ mass typically al-ludes to a heightened degree of maturation. Accelerated organ development leads to a smoother coordination of internal organs during vital biological processes like oxygen transport, blood circulation, lipid metabolism, and digestion. This refinement of these processes can positively impact growth and economic traits.” Please see page 2 Lines 46-52.
  • We have changed sentences “Therefore, understanding the genetic architecture of heart weight (Heart WT), liver weight (Liver WT), spleen weight (Spleen WT), lung weight (Lung WT), kidney weight (Kidney WT), and stomach weight (Stomach WT) will advance the genetic progress and improve the value of economic traits which further contribute to the effective implementation of breeding programs.” to “Thus, comprehending the genetic architecture of heart weight (Heart WT), liver weight (Liver WT), spleen weight (Spleen WT), lung weight (Lung WT), kidney weight (Kidney WT), and stomach weight (Stomach WT) will propel genetic progress and facilitate the successful implementation of breeding programs.”. Please see page 2 Lines 57-61.
  • We add a sentence “Integrating SNP results from GWAS as a source of prior biological information in the improvement program enhances the selection process by assigning higher weight to key SNPs that are critical for improving internal organ weight traits.” in the revised manuscript. Please see page 3 Lines 103-106.

M&M

Line 103: Correct sentence. “Test for normal distribution…”

Response: Thanks for your comment. We have changed the sentence “Test for normal distribution of descriptive statistics of internal organ traits using R software.”to”R software was used to test the normal distribution of descriptive statistics of internal organ traits.”. Please see Page 3 Lines 126-127.

Comments on section 2.2: It is important to describe how these measurements were taken. The procedures.

Response: Thanks for your comment. We have revised the manuscript following the reviewer’s comments. In the revised manuscript, we add the some information, i.e.,: "Experimental animals were selected from a DLY three-way crossbred commercial line with no overlapping blood relations, through random selection based on genealogy., in which 89 Duroc boars were mated with 397 Landrace × Yorkshire sows to produce a large number of offspring. All pigs were raised in four farms of Guangdong Wens Food Group Co., Ltd (Guangzhou, China). In brief, a total of 1518 individuals (757 boars and 764 sows) were reared with free access to water and feed and were fattened to 115 kg. They were euthanized in 13 batches with a 24-hour interval between each batch and had an average slaughter age of about 7 months. After the pigs were euthanized, their phenotypes were recorded and their internal organs were excised, emptied, flushed, blotted dry, and weighed immediately using an electronic scale with a range of 0.0 kg to 300 kg and accuracy of ± 100 g. The scale was calibrated using the linear calibration method with 20%MAX or 60%MAX weight. The organ distribution is shown in Figure 1. R software was used to test the normal distribution of the descriptive statistics of the internal organ traits.”Please see page 3 Line 113-127.

Lines 114-118: replace “with standard:” for “with the following parameters:” Replace “filtered” for “excluded”. Exclude “still”. Replace “single-trait GWAS and multi-trait GWAS” for “single-trait and multi-trait GWAS”.

Response: Thank you for this importantsuggestions. The text has been modified according to the reviewer’s comment, please see Page 4 Lines 139-143 in the revised manuscript.

Lines 138-142: Provide more details on the multivariate linear models or specify the one that the authors chose to conduct the analysis.

Response: We are very grateful to the reviewer for suggestions. According to the reviewer’s suggestions, we have added a few sentences in Page 5 Lines 169-178 to provide more details about the multivariate linear mixed models. “The multivariate linear mixed models was as follows:

where  is a matrix of six internal organs for 1518 individuals;  is a covariable matrix (fixed effects);  is a matrix of the corresponding coefficients;  is a vector that marks the genotypes;  is a vector of marker effect sizesfor six internal organs weight. U denotes the random effects; E is a matrix of errors;  denotes the kinship matrix;  denotes symmetric matrix of genetic variance component; I is a identity matrix; denotes symmetric matrix of environmental variance component;  denotes the n×d matrix normal distribution with mean 0; denotes row covariance matrix;  denotes column covariance matrix.” in the revised manuscript.

Line 144: number of false negative results or false positive results? I believe false positive is more appropriate.

Response: We are sorry for the ambiguous described. We have improved this sentences to ”As the Bonferroni correction can lead to an over-correction and can be too conservative, this can result in a limited number of labeled association P-values that meet the standard across the genome. This can lead to a high false negative rate. To address this issue, the False Discovery Rate (FDR) is employed as a correction to the threshold.”. Please see Page 5 Lines 179-183 in the revision.

Line 152: “MAF values > 0.05”; “P-values < 10³”. Check parameters because the authors were stricter in the genotyping quality control.

Response: We are sorry for the ambiguous described. The three parameters of "MAF values > 0.05, Mendelian errors < 2, and P-values < 10-3" are default parameters in Haploview v4.2 software. In our study, due to more stringent quality control standards, these three parameters did not affect the results of haplotype block analysis. We have modified this sentence to “Using the default parameters of Haploview 4.2 [22] (MAF > 0.05, Mendelian error < 2, and P-value < 10-3 for the HWE test) to define the linkage disequilibrium (LD) blocks of SNPs.” Please see Page 5 Line192-194.

Line 160: “victor”. Typo.

Response: Corrected

Section 2.6: Describe how the SNP-based heritability was computed. Provide a reference for that as well.

Response: Thank you for your comment. We have modified sentence to ”In present study, the restricted maximum likelihood (REML) method was used to assess SNP-based heritability of each internal organ weight trait and calculated the percentage of phenotypic variation that could be explained by significant SNPs by GCTA software. SNP-based heritability and the percentage of phenotypic variation explained by significant SNPs were calculated as follows[23]: ”. According to the reviewer's suggestion, we provide a reference for SNP-based heritability. Please see page 5 Lines 198-203.

Line 171: “in involved”. Text correction need.

Response: Corrected

Section 2.7: “nearest peak SNPs”. What was the window size established? Specify that in text. Usually we use biomaRt for that, but it seems the authors went to a manual search, right? I have found information in the Results section, but I believe this search and why the use of 200kb should be better explained in the M&M.

Response: We are sorry for the ambiguous description. We have modified the sentences to “Our previous studies on this population showed that the average r2 of 0.2 is about 200 kb apart [24], the range of searching for the functional gene closest to the position of the significant SNP is based on the LD decay distance (r2 = 0.2) of the populations [25]. We used the "biomaRt" package [26] in R, based on the Sus scrofa 11.1 genome version database (http://ensemble.org/Sus_scrofa/Info/Index, accessed September 20, 2022). Genes nearest the significant SNPs are list in the Table 3. We conducted a search of both PubMed and the relevant literature to examine the correlation between the nearest peak SNPs of all the candidate genes and the internal organ weight traits be-ing analyzed.” Please see page 6 Lines 211-219.

Results and Discussion

Lines 175-180. Text edition suggestion: The descriptive phenotypic statistics and estimated heritabilities (h²) for analyzed of the internal organ weights are listed in Table 1.

Response: We thank the reviewer for this suggestion. We have changed the sentence to “The descriptive phenotypic statistics and estimated heritabilities (h2) for analyzed of the internal organ weights are listed in Table 1.” in the revised. Please see page 6 Line 227.

Second sentence needs correction. Please correct it. “For internal organ weights, which were important parameter responding organ function and the level of individual development.

Response: We have correced the sentence to “The weight of internal organs is a crucial indicator of internal organ development and has a significant impact on organ function.”. Please see page 6 Lines 228-230.

Text edition suggestion: The average weight of heart WT, liver WT,spleen WT, lung WT, kidney WT and stomach WT in DLY pigs were 455.57g, 1763.61g, 212.54g, 1020.54g, 0.41kg, and 727.68g, respectively.

Response: The text has been modified according to the reviewer’s suggestion. Please see page 6 Lines 231-233.

Exclude the next “for DLY pigs” in line 181.

Response: The text has been modified according to the reviewer’s suggestion. Please see page 6 Line 234.

Line 181: replace “had moderate” for “had moderate to high”.

Response: The text has been modified according to the reviewer’s suggestion. Please see page 6 Line 235.

Line 184: replace “were also moderate heritability” for “were moderate to high estimations”.

Response: The text has been modified according to the reviewer’s suggestion. Please see page 6 Line 238.

Line 195: result -> results

Response: Corrected.

Line 195-200: “The results… improve the organ weight traits”. These is a long sentence. Try to break it down into two or more sentences.

Response: Thanks for your helpful suggestion. Combining your next two suggestions, we have rewritten this sentences to “The results revealed moderate to low genetic correlations among the six internal organ weight traits. Heart WT had moderate genetic correlations with liver WT, lung WT, and kidney WT, suggesting that these traits could be improved together in pig breeding programs. On the other hand, stomach WT showed close to 0 genetic correlations with most of the other traits, indicating stomach WT traits are less influenced by other traits when they are inherited. Therefore, reasonable breeding strategies need to be designed to improve internal organ weight traits. The phenotypic correlation results showed that the correlation coefficients between the phenotypes were at moderate to high levels, excluding the low phenotypic correlation coefficients between Lung WT and Liver WT, Spleen WT and Kidney WT, especially the phenotypic correlation coefficients of liver WT and kidney WT were as high as 0.62. When selecting for a certain phenotype in pig breeding, it is advantageous to also consider other related traits.”in the revised manuscript. Please see pages 6-7 Lines 250-261.

Line 197: What does a “significant genetic independence” mean? The term significant is not appropriate here.

Response: We thank the reviewer for this reminding. Here, "significant genetic independence" means that the six internal organ weight traits are less influenced by other traits when inherited. We have modified the “significant genetic independence” to “stomach WT traits are less influenced by other traits when they are inherited” in the revised manuscript. We have excluded the “significant” in the revised manuscript. Please see page 7 Lines 254-255.

Line 196: “generally low”. Some moderate correlation coefficients were estimated. There is room for discussing the correlation results better. Value 0.62 for liver WT and kidney WT stood out.

Response: Thanks for your helpful suggestions. Combine the two suggestions above, we have re-discussed the phenotypic correlation and genetic correlation results. Please see pages 6-7 Lines 250-261.

Line 211: Please, show Q-Q plots in the manuscript or in the supplementary file. Now I see Fig 2 should be refereed in this section.

Response: Thank you for your suggestion. We have refereed Figure 2 in this section, “Q-Q plots were conducted for heart WT, liver WT, spleen WT, lung WT, kidney WT and stomach WT to further assess population stratification (together with the Manhattan plots: Figures 2).”. Please see page 7 Line 278.

Line 227: and nearby -> and is located nearby. What is the distance between SNP 226 WU_10.2_12_6703865 and gene CD300LB in bp? Include this information in the manuscript.

Response: Thank you for your suggestion. We have modified this sentence to “The most significant SNP, WU_10.2_12_6703865 on SSC12, explains 1.70% of the phenotypic variation and is about 44 kb downstream of the CD300LB gene.” in the revised. Please see page 7 Line 297.

Line 235: and within -> and is located within

Response: corrected

Line 246: variation is -> variation and is

Response: corrected

Line 254: that both vs may both play. Please, verify the text.

Response: Thank you for your comment. We have changed the sentence to “SNP name ALGA0110225 explained 1.81% of the phenotypic variance and is located 97 kb downstream of PBX3, indicating PBX3 and ALGA0110225 may both play a role in the Lung WT.” in the revised. Please see page 8 Lines 324-326.

Line 282: reported by pigQTL database-> reported QTL documented in the pigQTL database. Include reference for pigQTL database.

Response: The text has been modified according to the reviewer’s suggestion. We have added the reference for pigQTL database, “The above GWAS results showed that none of the SNPs associated with internal organ weight overlapped with previously reported QTL documented in the pigQTL database [4].” Please see page 9 Line 355.

Figure 2: replace “Manhattan and Q-Q plots of internal organ weight traits” for “Manhattan and Q-Q plots of internal organ weight traits in the single-trait GWAS”.

Response: The text has been modified according to the reviewer’s suggestion. Please see page 11 Line 364.

Table 3: Sort column SSC within trait to organize this table.  OR sort the table by the column “p-value”  or PEV from the highest to the lowest value. The same works for Table 4.

Response: Thanks for your helpful suggestion. We have sorted each trait in Tables 2 and 4 by PEV column from highest to lowest. Please see page 11 Line 373 (Table 3) and page 13 Line 432 (Table 4).

3.4. Multi-trait GWAS

Lines 312-315: Specify methodology for multi-trait GWAS in M&M and cite the reference.

Response: Thank you for your suggestion. We have modified the sentence to " Moreover, the GEMMA software [17] was used to implement the multivariate linear mixed models (mvLMM) [18] for multi-trait GWAS to assess pleiotropic SNPs. The mvLMM and LMM models were both implemented as described in previous studies [19]." in M&M. The multivariate linear mixed model (mvLMM) method were used for multi-trait GWAS, and we have provided more detailed information about the mvLMM in the answer to question “Lines 138-142”, and the text has cited references as suggested by the reviewer. Please see page 5 Lines 165-178.

Line 337: “internal organ-heavy traits”?

Response: corrected

Line 316: “Populus trichocarpa” in italic.

Response: corrected

Line 321: “All these results indicated that multi-trait GWAS supplemented the 321 results of single-trait GWAS and improved the statistical effect.” I believe this statement can be explained more because of the methodology implemented than the results found.

Response: Thank you for your helpful suggestion. We have modified sentence to “In order to improve the statistical effect, multi-trait GWAS was performed separately for each SNPs individually by com-bining the joint analysis of six internal organ weight traits. ”. Please see page 12 Lines 387-388.

Conclusion

Replace “GWASs” with GWAS

Response: corrected

Round 2

Reviewer 1 Report

Thank you to the authors for their changes. With these changes, I feel the paper is ready to be published.